# Online Meta-Learning via Learning with Layer-Distributed Memory

**Sudarshan Babu**
TTI-C
sudarshan@ttic.edu

**Pedro Savarese**
TTI-C
savarese@ttic.edu

**Michael Maire**
University of Chicago
mmaire@uchicago.edu

## Abstract

We demonstrate that efficient meta-learning can be achieved via end-to-end training of deep neural networks with memory distributed across layers. The persistent state of this memory assumes the entire burden of guiding task adaptation. Moreover, its distributed nature is instrumental in orchestrating adaptation. Ablation experiments demonstrate that providing relevant feedback to memory units distributed across the depth of the network enables them to guide adaptation throughout the entire network. Our results show that this is a successful strategy for simplifying meta-learning – often cast as a bi-level optimization problem – to standard end-to-end training, while outperforming gradient-based, prototype-based, and other memory-based meta-learning strategies. Additionally, our adaptation strategy naturally handles online learning scenarios with a significant delay between observing a sample and its corresponding label – a setting in which other approaches struggle. Adaptation via distributed memory is effective across a wide range of learning tasks, ranging from classification to online few-shot semantic segmentation.

## 1   Introduction

Meta-learning or learning-to-learn is a paradigm that enables models to generalize to a distribution of tasks rather than specialize to just one task [1, 2]. When encountering examples from a new task, we would like the model to adapt to the new task after seeing just a few samples. This is commonly achieved via episodic training of deep neural networks, where, in each episode, the network is exposed to a variety of inputs from the same distribution [3, 4], and the distribution shifts over episodes. The ability of deep networks to adapt to a new task within just a few samples or iterations is central to the application of meta-learning methods in few-shot and online learning scenarios [5, 6].

A recent surge of interest directed towards meta-learning using neural networks has spurred development of a variety of methods [7–9]. In a standard episodic training framework, a network must adapt to a sampled task (or collection of tasks) and incurs a generalization loss for that task (or collection); this generalization loss is backpropagated to update the network weights. Methods differ in the underlying architecture and mechanisms they use to support adaptation. Strategies include using gradient descent in an inner loop, storing and updating prototypes, parameterizing update rules by another neural network, and employing neural memory [3, 10–12]. Section 2 provides an overview.

We focus on memory-based meta-learning, and specifically investigate the organization of neural memory for meta-learning. Motivating this focus is the generality and flexibility of memory-based approaches. Relying on memory for adaptation allows one to cast meta-learning as merely a learning problem using a straightforward loss formulation (viewing entire episodes as examples) and standard optimization techniques. The actual burden of adaptation becomes an implicit responsibility of the memory subsystem: the network must learn to use its persistent memory in a manner that facilitates task adaptation. This contrasts with explicit adaptation mechanisms such as stored prototypes.

35th Conference on Neural Information Processing Systems (NeurIPS 2021).

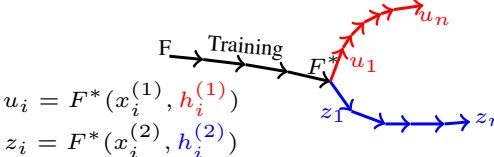

$$u_i = F^*(x_i^{(1)}, h_i^{(1)})$$
$$z_i = F^*(x_i^{(2)}, h_i^{(2)})$$

Figure 1: **Adaptation in activation space.** A trained memory-based model $F^*$ adapting to two different tasks (red and blue path) using the corresponding persistent states $h_i^1$ and $h_i^2$ at the $i^{th}$ time step of both tasks. $x_i^{(1)}$ and $x_i^{(2)}$ are samples of task 1 (red) and 2 (blue) at time step $i$.

In this implicit adaptation setting, memory architecture plays a crucial role in determining what kind of adaptation can be learned. We experimentally evaluate the effectiveness of alternative neural memory architectures for meta-learning and observe particular advantages to distributing memory throughout a network. More specifically, we view the generic LSTM equations, $Wx + Wh_{-1}$, as adaptation induced by hidden states in activation space (see Figure 1). By distributing LSTM memory cells across the depth of the network, each layer is tasked with generating hidden states that are useful for adaptation. Such a memory organization is compatible with many standard networks, including CNNs, and can be achieved by merely swapping LSTM memory cells in place of existing filters.

Our simple approach also contrasts with several existing memory-based meta-learning methods used in both generative and classification tasks [13–17]. These methods view memory as a means to store and retrieve useful inductive biases for task adaptation, and hence focus on designing better read and write protocols. They typically have a feature extractor that feeds into a memory network that performs adaptation, whereas our architecture makes no such distinction between stages.

We test the efficacy of network architectures with distributed memory cells on online few-shot and continual learning tasks as in Santoro et al. [13], Ren et al. [18] and Javed and White [6]. The online setting is challenging for two reasons: 1) It is empirically observed that networks are not well suited for training/adaptation with a batch size of one [19]; 2) In this setting the model has to adapt to one image at a time step, thus having to deal with a prolonged adaptation phase. For these reasons, we see these tasks as suitable for evaluating the adaptation capabilities of the hidden states generated by the network.

We empirically observe that our method outperforms strong gradient-based and prototypical baselines, delineating the efficacy of the local adaptation rule learnt by each layer. Particularly important is the distributed nature of our memory, which allows every network layer to adapt when provided with label information; in comparison, restricting adaptability to only later network layers delivers far less compelling performance. These results suggest that co-design of memory architecture and meta-training strategies should be a primary consideration in the ongoing development of memory-based meta-learning. We further test our model in a harder online few-shot learning scenario, wherein the corresponding label to a sample arrives after a long delay [20]. Our method adapts seamlessly, without requiring any changes to the model, while, in this setting, other adaptation strategies struggle. These results highlight promising directions for advancing and simplifying meta-learning by relying upon distributed memory for adaptation.

## 2    Related Work

Early work on meta-learning introduces many relevant concepts. Schmidhuber [21] proposes using task specific weights, called fast weights, and weights that are adapted across tasks, called slow weights. Bengio et al. [2] updates the network via a learning rule which is parameterized by another neural network. Thrun [22] presents meta-learning in a life-long scenario, where the algorithm accrues information from the past experiences to adapt effectively for the task at hand. Hochreiter et al. [23] train a memory network to learn its own adaptation rule via just its recurrent states. These high level concepts can be seen in more recent methods. We group current meta-learning methods based on the nature of adaptation strategy and discuss them below.

**Gradient-based Adaptation Methods.** Methods that adapt via gradients constitute a prominent class of meta-learning algorithms [9]. Model-agnostic meta-learning (MAML) [4] learns an initialization that can efficiently be adapted by gradient descent for a new task. Finn et al. [24] focus on learning a network that can use experience from previously seen tasks for current task adaptation. They adapt to the current task by using a network that is MAML pre-trained on the samples from the previous task. Nagabandi et al. [25], Caccia et al. [26] perform online adaptation under non-stationary distributions, either by using a mixture model or by spawning a MAML pre-trained network when the

input distribution changes. Javed and White [6], Beaulieu et al. [27] employ a bi-level optimization routine similar to MAML, except the outer loop loss is catastrophic forgetting. They thereby learn representations that are robust to forgetting and accelerate future learning under online updates.

**Memory and Gradient-based Adaptation.** Andrychowicz et al. [28], Ravi and Larochelle [10] learn an update rule for network weights by transforming gradients via a LSTM, which outperforms human-designed and fixed SGD update rules. Munkhdalai and Yu [29] learn a transform that maps gradients to fast (task specific) weights, which are stored and retrieved via attention during evaluation. They update slow weights (across task weights) at the end of each task.

**Prototypical Methods.** These methods learn an encoder which projects training data to a metric space, and obtain class-wise prototypes via averaging representations within the same class. Following this, test data is mapped to the same metric space, wherein classification is achieved via a simple rule (*e.g.,* nearest neighbor prototype based on either euclidean distance or cosine similarity) [5, 30, 31]. These methods are naturally amenable for online learning as class-wise prototypes can be updated in an online manner as shown by Ren et al. [18].

**Memory-based Adaptation.** Santoro et al. [13] design efficient read and write protocols for a Neural Turning Machine [32] for the purposes of online few-shot learning. Rae et al. [33] design sparse read and write operations, thereby making them scalable in both time and space. Ramalho and Garnelo [11] use logits generated by the model to decide if a certain sample is written to neural memory. Mishra et al. [7] employ an attention-based mechanism to perform adaptation, and use a CNN to generate features for the attention mechanism. Their model requires storing samples across all time steps explicitly, thereby violating the online learning assumption of being able to access each sample only once. All of these methods mainly focus on designing better memory modules either via using more recent attention mechanisms or by designing better read and write rules to neural memory. These methods typically use a CNN which is not adapted for the current task. Our approach differs from these methods, in that we study efficient organization of memory for both online few-shot learning and meta-learning more generally, and show that as a consequence of our distributed memory organization, the entire network is capable of effective adaptation when provided with relevant feedback.

Kirsch and Schmidhuber [34] introduce an interesting form of weight sharing wherein LSTM cells (with tied weights) are distributed throughout the width and depth of the network, however each position has its own hidden state. Further, they have backward connections from the later layers to the earlier layers, enabling the network to implement its own learning algorithm or clone a human-designed learning algorithm such as backprop. Both our model and theirs implement an adaptation strategy purely using the recurrent states. The difference, however, is in the nature of the adaptation strategy implemented in the recurrent states. Similar to conventional learning algorithms, their backward connections help propagate error from the last layer to the earlier layers. In our architecture, the feedback signal is presented as another input, propagated from the first layer to the last layer.

In addition to being used in classification settings, Guez et al. [35] employ memory-based meta-learning approach to perform adaptation for reinforcement learning tasks indicating the generality of using memory as a means for adaptation.

**Few-shot Semantic Segmentation.** Few-shot segmentation methods commonly rely on using prototypes [36, 37], though recent approaches include gradient-based methods analogous to MAML [38]. The methods that use neural memory typically employ it in final network stages to fuse features of different formats for efficient segmentation: Li et al. [39] use ConvLSTMs [40] to fuse features from different stages of the network; Valipour et al. [41] to fuse spatio-temporal features while segmenting videos; Hu et al. [42] use a ConvLSTM to fuse features of query with the features of support set; Azad et al. [43] use a bidirectional ConvLSTM to fuse segmentation derived from multiple scale space representations. We differ from these works in organization, use of, and information provided to memory module: 1) Memory is distributed across the network as the sole driver of adaptation; 2) Label information is provided to assist with adaptation.

**Meta-learning Benchmarks.** Caccia et al. [26] present benchmarks that measure the ability of a model to adapt to a new task, using the inductive biases that it has acquired over solving previously seen tasks. More specifically, the benchmark presents an online non-stationary stream of tasks, and the model's ability to adapt to a new task at each time step is evaluated. Note that they do not measure

the model's ability to remember earlier tasks; they only want the model to adapt well on a newly presented task.

Antoniou et al. [44] present benchmarks for continual few-shot learning. The network is presented a number of few-shot tasks, one after the other, and then is expected to generalize even to the previously seen tasks. This is a challenging and interesting setup, in that, the network has to show robustness to catastrophic forgetting while learning from limited data. However, we are interested in evaluating the online adaptation ability of models, while Antoniou et al. [44] feed data in a batch setting. We follow experimental setup as in Javed and White [6], where in, the model is required to remember inductive biases acquired over a longer time frame when compared to the experimental setup used by Antoniou et al. [44].

## 3  Methodology

### 3.1  Problem: Online Few-shot Learning

This setting combines facets of online and few-shot learning: the model is expected to make predictions on a stream of input samples, while it sees only a few samples per class in the given input stream. In particular, we use a task protocol similar to Santoro et al. [13]. At time step $i$, an image $x_i$ is presented to the model and it makes a prediction for $x_i$. In the following time step, the correct label $y_i$ is revealed to the model. The model's performance depends on the correctness of its prediction at each time step. The following ordered set constitutes a task: $\mathcal{T} = ((x_1, null), (x_2, y_1), \cdots (x_t, y_{t-1}))$. Here $null$ indicates that no label is passed at the first time step, and $t$ is the total number of time steps (length) of the task. For a k-way N-shot task $t = k \times N$. The entire duration of the task is considered as the adaptation phase, as with every time step the model gets a new sample and must adapt on it to improve its understanding of class concepts.

### 3.2  Memory as Adaptation in Activation Space

Consider modulating the output of a network $F$ for input $x$ with a persistent state $h$: $u = F(x, h)$. Now, if adding $h$ aids in realizing a better representation $u$ than otherwise ($F(x)$), we could view this as adaptation in activation space. In Figure 1, model $F^*$ adapts to tasks using its persistent states $h$. Specifically let us consider the generic LSTM equations $Wx + W'h_{-1}$, we could view $Wx$ as the original response and $W'h_{-1}$ as modulation by a persistent state (memory) in the activation space. So, for the online learning task at hand, we seek to train a LSTM which learns to generate hidden state $h_i$ at each time step $i$, such that it could enable better adaptation in ensuing time steps. We note that adaptation in activation space has been discussed in earlier works. We use this perspective to organize memory better and to enable effective layer-wise adaptation across the network.

### 3.3  Model

**Architecture.** We distribute memory across the layers of the network, in order to enable the layers to learn local layer-wise adaptation rules. In particular, we use a model in which each layer of the feature extractor is a convolutional LSTM (CL) [40] followed by a LSTM [45] and a classifier, as shown in Figure 2.

Similar to the LSTM, each convolutional LSTM (CL) layer consists of its own input, forget, and output gates. The key difference is that convolution operations (denoted by $*$) replace matrix-vector multiplication. In this setup, we view the addition by $W_{hi} * h_{t-1}$ as adaptation in the $i_{th}$ time step within the input gate. The same view could be extended to other gates as well. The cell and hidden state generation are likewise similar to LSTM, but use convolution operations:

$$i_t = \sigma(W_{ii} * x_t + W_{hi} * h_{t-1}) \quad (1) \qquad c_t = f_t \odot c_{t-1} + i_t \odot \tanh(W_{ig} * x_t + W_{hg} * h_{t-1})$$
$$f_t = \sigma(W_{if} * x_t + W_{hf} * h_{t-1}) \quad (2) \tag{4}$$
$$o_t = \sigma(W_{io} * x_t + W_{ho} * h_{t-1}) \quad (3) \qquad h_t = o_t \odot \tanh(c_t) \tag{5}$$

In initial experiments, we observe that for tasks with 50 time steps these models did not train well. We hypothesize that this could be due to the same network being repeated 50 times, thereby inducing

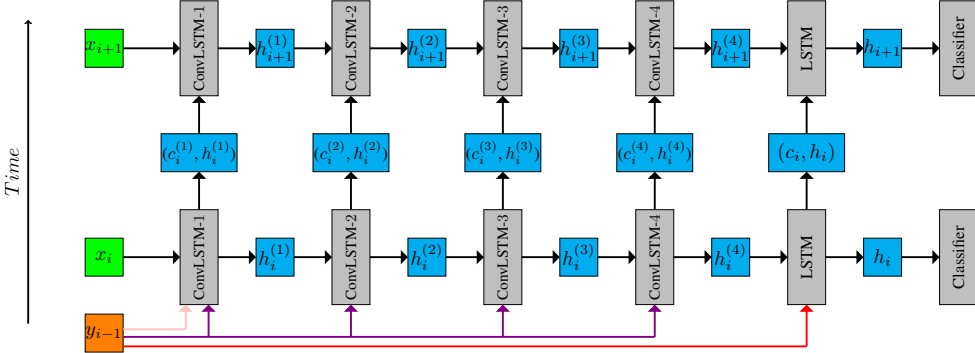

Figure 2: **Methodology.** An example distributed memory architecture consists of four layers of convolutional LSTMs (gray), followed by an LSTM (gray), and a classifier (gray). At the $i_{th}$ time step, the sample $x_i$ (green) and the previous sample's label $y_{-1}$ are presented to network. Three different label injection modes are shown. Pink: label is fed to the first CL layer only; Violet: label is fed to each CL layer; Red: label is fed to the LSTM layer. Label information is provided at every time step; for sake of clarity we avoid showing it at time $i + 1$ here. $h_i^{(j)}$, $c_i^{(j)}$ are the hidden and cell state of the $j_{th}$ layer at the $i_{th}$ time step. We use cyan, gray, orange, and green to denote persistent states, network parameters, label information, and input sample, respectively. Best viewed in color.

an effectively very deep network. We resolve this issue by adding skip connections between the second layer and the fourth layer (omitted in Figure 2). Further discussion on this is in Appendix B.

**Label Encoding.** As label information is essential for learning an adaptation rule, we inject labels offset by one time step to the ConvLSTM feature extractor and the LSTM. This provides the opportunity for each layer to learn an adaptation rule. For a k-way classification problem involving images of spatial resolution $s$, we feed the label information as a $k \times s^2$ matrix with all ones in the $c^{th}$ row if $c$ is indeed the true label. We reshape this matrix as a $k \times s \times s$ tensor and concatenate it along the channel dimension of image at the next time step. To the LSTM layer, we feed the label in its one-hot form by concatenating it with the flattened activations from the previous layer.

### 3.4 Training and Evaluation

Following Santoro et al. [13], we perform episodic training by exposing the model to a variety of tasks from the training distribution $\mathcal{P}(\mathcal{T}_{train})$. For a given task, the model incurs a loss $\mathcal{L}_i$ at every time step of the task; we sum these losses and backpropagate through the sum at the end of the task. This is detailed in Algorithm 1 in Appendix A. We evaluate the model using a partition of the dataset that is class-wise disjoint from the training partition. The model makes a prediction at every time step and adapts to the sequence by using its own hidden states, thereby not requiring any gradient information for adaptation. Algorithm 2 in Appendix A provides details.

## 4 Experiments

### 4.1 Online Few-Shot Learning

We use CIFAR-FS [46] and Omniglot [47] datasets for our few-shot learning tasks; see Appendix A for details. We adopt the following methods to serve as baselines for comparison.

**LSTM and NTM.** Santoro et al. [13] use a LSTM and a NTM [32] with read and write protocols for the task of online few shot learning. Both aim to meta-learn tasks by employing a neural memory.

**Adaptive Posterior Learning (APL).** Ramalho and Garnelo [11] propose a memory-augmented model that stores data point embeddings based on a measure of *surprise*, which is computed by the loss incurred by each sample. During inference, they retrieve a fixed number of nearest-neighbor data embeddings, which are then fed to a classifier alongside the current sample.

**Online Prototypical Networks (OPN).** Ren et al. [18] extend prototypical networks to the online case, where they sequentially update the current class-wise prototypes using weighted averaging.

**Contextual Prototypical Memory (CPM).** Ren et al. [18] improve on OPN by learning a representation space that is conditioned on the current task. Furthermore, weights used to update prototypes are determined by a newly-introduced gating mechanism.

Table 1 shows that our model outperforms the baselines in most settings. These results suggest that the adaptation rules emergent from our design are more efficient than adaptation via prototypes, and adaptation via other memory-based architectures. In the CIFAR-FS experiments, the prototypical methods outperform our method only in the 1-shot scenario. As the 5-shot and 8-shot scenarios have a longer fine-tuning or adaptation phase, this shows that our method is more adept at handling tasks with longer adaptation phases. One reason could be that the stored prototypes which form the persistent state of OPN and CPM are more rigid than the persistent state of our method. The rigidity stems from the predetermined representation size of each prototype, which thereby prevents allocation of representation size depending upon classification difficulty. In our architecture, the network has the freedom to allocate representation size for each class as it deems fit. Consequently, this may help the network learn more efficient adaptation strategies that improve with time.

We examine the importance of distributed adaptation through ablation experiments that vary the layer into which we inject label information. Table 2 shows that models with feature extractors that do not receive label information are outperformed by the model whose earlier layers do receive label information (injecting into CL-1); the latter is even better than pre-trained models. By distributing memory across each layer and allowing label information to flow to each memory module, we enable every layer to learn its own adaptation rule. Here, the CNN baselines are pre-trained with MAML; these pre-trained networks replace the ConvLSTM part and are jointly trained with the LSTM (which receives the labels) and classifier. In these cases, we just replace the ConvLSTM in Figure 2 with a CNN. During the meta-testing phase, the CNN is a just feature extractor and the burden of adaptation falls entirely on the LSTM. In CNN-F, we freeze the weights during meta-training. Our CL+LSTM, restricted to adapt only in the final layer (3rd row; label injection into final LSTM layer only) performs comparably to the CNN baselines. The same model, with full adaptivity (last row) outperforms.

## 4.2 Delayed Feedback

We consider a task similar to online few-shot classification (Section 3.1), except instead of offset by one timestep, labels are offset by a delay parameter. Supposing the label delay is 3, then the task $\mathcal{T}$ is presented to the model as the sequence: $\mathcal{T} = \big((x_1, null), (x_2, null), (x_3, null), (x_4, y_1), \cdots (x_t, y_{t-3})\big)$, where $t$ is the sequence length. The model must discern and account for the time delay.

Table 3 shows that our network can learn under these conditions, though performance decreases with increase in delay. This could be imputed to difficulty in associating the hidden representation of a sample with the correct label, consequently creating a noisy environment for learning adaptation rules. We see that pre-training helps: we take our network pre-trained for label delay of 1 and meta-train for tasks with label delay of 5. This improves the model accuracy to outperform the model directly trained with label delay of 4. This could be because the necessary adaptation rules are already learnt by the pre-trained model, and it only has to learn the quantum of delay. Furthermore, from Tables 1 and 3, even with a delay of 2 our method outperforms CPM with no delay.

| Model | Omniglot | | | CIFAR-FS | | |
|---|---|---|---|---|---|---|
| | 1-shot | 5-shot | 8-shot | 1-shot | 5-shot | 8-shot |
| LSTM[†] | 85.3 (0.2) | 94.4 (0.1) | 95.8 (0.6) | - | - | - |
| NTM[†] | 88.7 (0.5) | 96.8 (0.1) | 97.3 (0.1) | - | - | - |
| APL | 89.1 (0.0) | 94.9 (0.0) | 95.7 (0.1) | 37.6 (0.6) | 45.8 (0.4) | 46.9 (0.7) |
| OPN | 91.2 (1.1) | 94.6 (1.1) | 95.8 (0.5) | 49.9 (0.2) | 54.9 (0.4) | 56.8 (0.1) |
| CPM | 94.5 (0.1) | 97.0 (0.1) | 97.4 (0.4) | **50.2** (0.2) | 55.8 (0.4) | 57.8 (0.4) |
| CL+LSTM | **96.8** (0.5) | **99.4** (0.2) | **99.7** (0.2) | 47.6 (1.0) | **56.7** (1.6) | **61.0** (1.3) |

Table 1: **Omniglot and CIFAR-FS results for 5-way online few-shot learning**. Shown are average and standard deviation across 3 runs. Methods with hand-designed memory mechanisms, like CPM and APL, benefit less from increased number of samples. Distributed memory in CL+LSTM comfortably outperforms other adaptation methods. [†] These methods fail to train on CIFAR-FS.

| Model | Pre-training | Label Injection Layer | 1-shot | 5-shot | 8-shot |
|-------|:---:|:---:|:---:|:---:|:---:|
| CNN-F + LSTM | ✓ | LSTM | 89.1 (0.2) | 96.4 (0.0) | 97.0 (0.2) |
| CNN + LSTM | ✓ | LSTM | 93.9 (0.1) | 97.9 (0.2) | 98.2 (0.4) |
| CL+LSTM | ✗ | LSTM | 94.7 (0.6) | 97.3 (0.7) | 97.9 (0.6) |
| CL+LSTM | ✗ | CL-1, LSTM | **96.8** (0.5) | **99.4** (0.2) | **99.7** (0.2) |

Table 2: **Ablation experiments on Omniglot 5-way online few-shot learning**. Results in (%) (average and deviation across 3 runs), comparing different feature extractors (pre-trained or not) while providing labels offset by 1 time step. Outputs of feature extractors are given to a LSTM (Figure 2). Our distributed memory model (CL+LSTM), with offset labels provided at the first layer, outperforms other models: since memory is distributed across layers, and since label information is provided to the entire network, each layer has sufficient information to learn its own adaptation rule.

In this setting, our model can be used in a seamless manner, without having to make any adjustments. Gradient-based and prototypical methods cannot be used as is, and would require storing the samples (violating online assumption) for the time period of delay, causing memory usage to grow linearly with delay; in contrast, it is constant for our method. Further, to use prototypical or gradient-based methods, we would have to know the delay parameter in advance; our network learns the delay.

## 4.3 Online Continual Learning

We address the problem of continual learning in the online setting. In this setup, the model sees a stream of samples from a non-stationary task distribution, and the model is expected to generalize well even while encountering samples from a previously seen task distribution. Concretely, for a single continual learning task we construct $n$ subtasks from an underlying dataset and first present to the model samples from the first subtask, then the second subtask, so on until the $n^{th}$ subtask in that order. Once the model is trained on all $n$ subtasks sequentially, it is expected to classify images from any of the subtasks, thereby demonstrating robustness catastrophic forgetting [48].

**Task Details.** We use the Omniglot dataset for our experiments. Following [6], we define each subtask as learning a single class concept. So in this protocol a single online 5-way 5-shot continual learning task is defined as the following ordered set: $\mathcal{T} = (\mathcal{T}_1, \mathcal{T}_2, \mathcal{T}_3, \mathcal{T}_4, \mathcal{T}_5)$. Here, subtask $\mathcal{T}_i$ contains 5 samples from 1 particular Omniglot class. After adaptation is done on these 5 subtasks (25 samples) we expect the model to classify samples from a query set consisting of samples from all of the subtasks. The performance of the model is the prediction accuracy on the query set. We experiment by varying the total number of subtasks from 5 to 20 as in Figure 3.

**Training Details.** We perform episodic training by exposing our model to a variety of continual learning tasks from the training partition. At the end of each continual learning task, the model incurs a loss on the query set. We update our model by backpropagating through this query set loss. Note that during evaluation on the query set, we freeze the persistent states of our model in order to prevent any information leak across the query set. Since propagating gradients across long time steps renders training difficult, we train our model using a simple curriculum of increasing task length every 5K episodes. This improves generalization and convergence. Appendix C presents more details. Further, we shuffle the labels across tasks in order to prevent the model from memorizing the training classes. During evaluation, we sample tasks from classes the model has not encountered. The model adapts to the subtasks using just the hidden states and then acquires the ability to predict on the query set, which contains samples from all of the subtasks. We use the same class wise disjoint train/test split as in Lake [47].

| Delay | Pre-training | 6-shot | 7-shot | 8-shot |
|:---:|:---:|:---:|:---:|:---:|
| 2 | ✗ | 56.8(0.1) | 58.4(0.2) | 60.0(0.0) |
| 4 | ✗ | 52.2(0.4) | 53.1(0.3) | 54.1(0.4) |
| 4 | ✓ | 56.1(0.6) | 57.6(0.3) | 59.0(0.1) |
| 5 | ✗ | 51.1(0.8) | 51.7(0.7) | 52.7(0.3) |
| 5 | ✓ | 55.2(0.3) | 56.4(0.0) | 57.4(0.4) |

Table 3: CIFAR-FS results (%) for 5-way online few-shot learning with delayed labels (average and deviation, in parentheses, across 2 runs). As expected, longer delays degrade the model's performance. Further, we observe that pre-training a model with 1-step delay improves the performance in tasks with 5-step and 4-step delays.

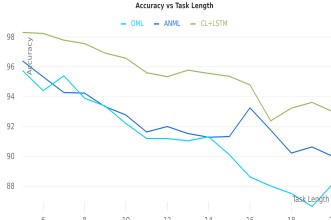

Figure 3: **Online few-shot continual learning.** Accuracy vs task length on Omniglot. CL+LSTM model all outperforms the baselines across tasks lengths, strongly suggesting that the CL+LSTM model is adept at storing inductive biases required to solve the subtasks within a given continual learning task.

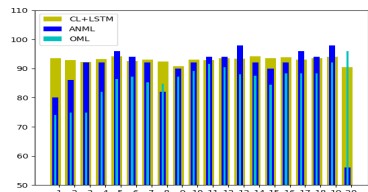

Figure 4: **Accuracy of subtasks within a 20 task continual learning task.** Typically in earlier tasks (first 10), CL+LSTM seems to have a higher accuracy than ANML and OML, suggesting that it is more immune to forgetting. ANML performs poorly in the $20^{th}$ task, about 56 % we impute that to momentum issues in training. Note: Excluding the $20^{th}$ task, the average ANML performance in the first 19 tasks is 91.78, while CL+LSTM averages 93.15 in the first 19 tasks.

**Baseline: Online Meta Learning (OML).** Javed and White [6] adopt a meta-training strategy similar to MAML. They adapt deeper layers in the inner loop for the current task, while updating the entire network in the outer loop, based on a loss measuring forgetting. For our OML experiments we use a 4-layer CNN followed by two fully connected layers. Appendix C provides implementation details.

**Baseline: A Neuromodulated Meta-Learning Algorithm (ANML).** Beaulieu et al. [27] use a hypernetwork to modulate the output of the trunk network. In the inner loop, the trunk network is adapted via gradient descent. In the outer loop, they update both the hypernetwork and the trunk network on a loss measuring forgetting. For our ANML experiments, we use a 4-layer CNN followed by a linear layer as the trunk network, with a 3-layer hypernetwork modulating the activations of the CNN. They use 3 times as many parameters as our CL+LSTM model. Appendix C provides details.

**Results.** Figure 3 plots average accuracy on increasing the length of the continual learning task. Task length is the number of subtasks within each continual learning task, which ranges from 5 to 20 subtasks in our experiments. As expected, we observe that the average accuracy generally decreases with increased task length for all models. However, the CL+LSTM model's performance degrades slower than the baselines, suggesting that the model has learnt an efficient way of storing inductive biases required to solve each of the subtasks effectively.

From Figure 4, we see that CL+LSTM is robust against forgetting, as the variance on performance across subtasks is low. This suggests that the CL+LSTM model learns adaptation rules that minimally interfere with other tasks.

**Analysis of Computational Cost.** During inference, our model does not require any gradient computation and fully relies on hidden states to perform adaptation. Consequently, it has lower computational requirements compared to gradient-based models – assuming adaptation is required at every time step. For a comparative case study, let us consider three models and their corresponding GFLOPs per forward pass: OML baseline (1.46 GFLOPs); CL+LSTM (0.40 GFLOPs); 4-layer CNN (0.30 GFLOPs) with parameter count similar to CL+LSTM. Here, we employ standard methodology for estimating of compute cost [49], with a forward and backward pass together incurring three times the operations in a forward pass alone.

We can extend these estimates to compute GFLOPs for the entire adaptation phase. Suppose we are adapting/updating our network on a task of length $t$ iterations. The OML baseline and the 4-layer CNN (adapting on gradient descent) would consume $4.38t$ GFLOPs and $0.9t$ GFLOPs, respectively. Our CL+LSTM model would consume only $0.40t$ GFLOPs; here, we drop the factor of three while computing GFLOPs for the CL+LSTM model, since we do not require any gradient computation for adaptation. During training, we lose this advantage since we perform backpropagation through time, making the computational cost similar to computing meta-gradients.

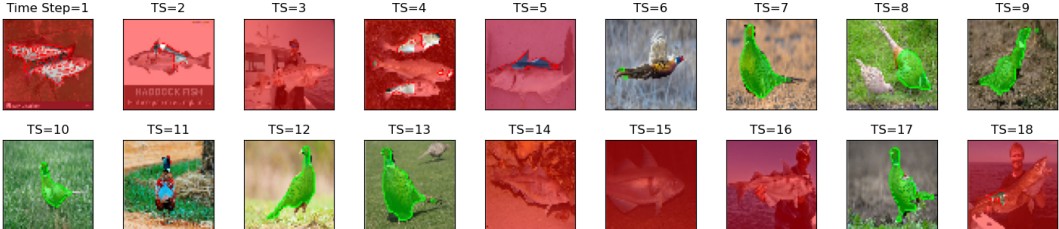

Figure 5: **Sample online few-shot segmentation task with distractors.** At each time step, the model gets one image, and its corresponding ground truth in the subsequent time step. The model is tasked with either segmenting or masking, depending on whether or not the image is a distractor. Here, the fish are distractors and the ducks are objects we want to segment. The portion highlighted in green is the predicted segmentation, and the red portion is the predicted mask. These are results from our 10-ConvLSTM model with label injection at the first layer.

## 4.4 Online few-shot Semantic Segmentation

These experiments investigate the efficacy and applicability of adaptation via persistent states to a challenging segmentation task and analyze the effectiveness of label injection for segmentation.

**Task Details.** We consider a binary segmentation task: we present the model a sequence of images, one at each time step (as in Figure 5), and the model must either segment or mask out the image based on whether it is a distractor. Similar to the classification tasks, we augment the ground truth segmentation information along the channel dimension. The ground truth is offset by 1 time step, so at the first time step we concatenate to the channel dimension an all -1 matrix as a null label, at the next time step we concatenate to the channel dimension the actual ground truth of the image at time step 1. If it is an image to be segmented, we concatenate the ground truth binary mask of the object and the background in the form of a binary matrix. If it is a distractor image, we concatenate to the channel dimension an all zeros matrix indicating that the entire image should be masked out. $k$-shot scores for segmentation is the IoU of the predicted segmentation on the $k + 1^{th}$ time the model sees the object we want to segment out. For $k$-shot masking scores, we compute the fraction of the object that has been masked, when model sees the distractor image for the $k + 1^{th}$ time step. We sample our episodes from the dataset FSS1000 [50]; more dataset details are in Appendix D.

The construction of this task avoids zero-shot transfer of inductive biases required for segmentation and forces the model to rely on the task data to learn which objects are to be segmented.

**Training Details.** We augment a 10 layer U-Net [51] like CNN with memory cells in each layer, by converting each convolution into a convolutional LSTM–referred to as CL U-Net (architecture details in Appendix D. We utilize episodic training, where each episode is an online few-shot segmentation task, as in Figure 5 with 18 time steps in total (9 segmentation images and 9 distractors). We follow a simple training curriculum to train: the first 100k episodes we train without any distractors; in the next 100k episodes we train with distractors as in Figure 5. Further training details are in Appendix D. The episodes presented during evaluation contain novel classes.

**Baselines.** We use a 10 layer U-Net like CNN pre-trained with MAML for segmentation without any distractors (architecture details in Appendix D). We use this model as our fine-tuning CNN baseline, in that we fine-tune the model on the online stream of images using gradient descent at each time step. From Table 4, we see that the model fails to mask out the distractors, indicating its inability to ability to adapt to the online feed.

From Table 4, we see that CL U-Net variants are capable of effective online adaptation; both models are capable of segmenting and masking images. However we observe that providing label information at the first layer significantly boosts our performance, thereby bolstering our claim that effective task adaptation can be achieved by providing relevant feedback to a network containing distributed memory.

## 4.5 Standard Supervised Learning

Finally, we assess whether our proposed model can be *directly* employed in a classic supervised learning setting *i.e.,* without requiring modifications in terms of architecture design. The central

| Model | Label Injection | Segmentation | | Masking | |
|---|---|---|---|---|---|
| | | 1-shot | 6-shot | 1-shot | 6-shot |
| Fine-tuned CNN | No | 65.3 | 64.0 | 23.1 | 23.6 |
| CL U-Net | 9th layer | 38.7 | 45.7 | 73.8 | 77.8 |
| CL U-Net | 1st layer | 49.2 | 56.0 | 84.6 | 91.7 |

Table 4: **Segmentation and Masking results on CL U-Net.** Fine-tuning a CNN fails to adapt to distractors, while CL U-Net variants demonstrate adept capacity for adaptation. Further label injection at the first layer outperforms label injection at the $9^{th}$ layer, suggesting that label injection enables network wide adaptation.

| Model | Params (M) | C10 | C100 |
|---|---|---|---|
| VGG-11 | 9.2 | 91.5 | 66.9 |
| CL-VGG-11 $(0.5\times)$ | 20.0 | 91.5 | 67.3 |
| CL-VGG-11 $(0.35\times)$ | 9.5 | 90.7 | 64.3 |
| ResNet-20 | 0.3 | 91.9 | 68.1 |
| CL-ResNet-20 $(0.5\times)$ | 0.6 | 90.9 | 63.2 |
| CL-ResNet-20 $(0.35\times)$ | 0.3 | 89.5 | 58.8 |

Table 5: **Test accuracy (%) of supervised learning on CIFAR.** Shown are averages across 3 runs. Parentheses following model names indicate shrinkage factor of filter sets. Variants perform comparably to original models, showing that they can be employed off-the-shelf in both meta and supervised learning settings.

motivation behind these experiments is to see if meta-learning methods can be applied to standard supervised learning tasks without requiring any change in methodology. Hence, in a setting when a priori knowledge of whether the task at hand is a standard supervised learning task or meta-learning task is unavailable, we could use ConvLSTM models. This is similar to the experiments done in [11], where they try to close the gap between standard supervised learning approaches and their meta-learning method applied to standard supervised learning tasks.

We use CIFAR data as our standard supervised learning benchmark [52]; further dataset details are in Appendix E. We use standard networks such as VGG [53] and ResNet [54] as our baselines. In Table 5, we observe that CL variants perform comparably in most cases. This affirms that the ConvLSTM model is capable of handling a conventional supervised learning scenario without any change in training procedure. Even in the absence of temporal signal, ConvLSTMs can still operate well. This is interesting since direct application of gradient-based meta-learners to the conventional supervised learning setting would require optimizing through a prohibitively long inner loop.

## 5 Conclusion

Our results highlight distributed memory architectures as a promising technical approach to recasting the problem of meta-learning as simply learning with memory-augmented models. This view has potential to eliminate the need for ad-hoc design of mechanisms or optimization procedures for task adaptation, replacing them with generic and general-purpose memory modules. Our ablation studies show the effectiveness of distributing memory throughout a deep neural network (resulting in an increased capacity for adaptation), rather than limiting it to a single layer or final classification stage.

We demonstrate that standard LSTM cells, when provided with relevant feedback, can act as a basic building block of a network designed for meta-learning. On a wide variety of tasks, a distributed memory architecture can learn adaptation strategies that outperform existing methods. The applicability of a purely memory-based network to online semantic segmentation points to the untapped versatility and efficacy of adaptation enabled by distributed persistent states.

## Acknowledgments and Disclosure of Funding

We thank Greg Shakhnarovich and Tri Huynh for useful comments. This work was supported in part by the University of Chicago CERES Center. The authors have no competing interests.

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
