# Appendices

## A  Details of Online Few-Shot Learning Experiments

### A.1  Datasets

**CIFAR-FS:** Bertinetto et al. [46] adapt the CIFAR-100 dataset for few-shot learning by performing a class-wise partition, yielding a training, validation, and testing set composed of 60, 16, and 20 classes, respectively. We sample 10-shot 5-way online few-shot tasks by sampling 5 classes and 10 samples per class (a total of 50 images), which are fed to the model at a rate of one image per time step. We report validation performance for all models.

**Omniglot:** This dataset has 1623 characters from 50 different alphabets, with 20 images per character [55]. Following Vinyals et al. [3], we resize images to $28 \times 28$ and augment the dataset by rotating each character by multiples of 90 degrees. We use the same split as Snell [56]. We report validation performance for all models.

### A.2  Task Details :

Figure 6 shows a sample online 10-shot 5-way task. As mentioned in Section 3.1, at time step $i$, we pass the $i_{th}$ sample and the label associated with the sample at the previous time step $i-1$. Further, at each time step the model predicts the label for the given sample; the model's efficacy is the accuracy of the predicted labels. Subsequently, in the next time step when the correct label is presented, we expect the model to update itself in order to achieve better generalization for future examples of the given task.

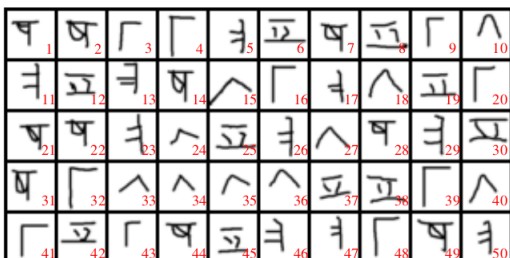

Figure 6: **A sample Omniglot-based online 10-shot 5-way task.** Each class consists of 10 examples and there are 5 classes in total. The red number at the bottom right of each character is the time-step at which that character is presented to the model.

### A.3  Training and Evaluation

---

**Algorithm 1:** Training model $\theta$ for online few shot learning

---

**TrainOnlineFewShot** $(\mathcal{P}(\mathcal{T}_{train}), \theta)$

 **inputs :** Task Distribution $\mathcal{P}(\mathcal{T}_{train})$;
     Parameter set $\theta$

 **while** *not converged* **do**
  Sample $\mathcal{T} \sim \mathcal{P}(\mathcal{T}_{train})$

  **foreach** $(x_i, y_{i-1}) \in \mathcal{T}$ **do**
   $\hat{y}_i = f_\theta(x_i, y_{i-1})$
   $\mathcal{L}_i = \mathcal{L}(\hat{y}_i, y_i)$ ; // $y_i$ is true label

  $\theta \leftarrow \theta - \nabla_\theta \sum_{i=1} \mathcal{L}_i$

 **return** $\theta$;

---

**Algorithm 2:** Evaluating model $\theta^*$ for online few shot learning

---

**TestOnlineFewShot** $(\mathcal{P}(\mathcal{T}_{test}), \theta^*)$

 **inputs :** Task Distribution $\mathcal{P}(\mathcal{T}_{test})$;
     Trained Parameter $\theta^*$

 **while** *not converged* **do**
  Sample $\mathcal{T} \sim \mathcal{P}(\mathcal{T}_{test})$

  **foreach** $(x_i, y_{i-1}) \in \mathcal{T}$ **do**
   $\hat{y}_i = f_\theta(x_i, y_{i-1})$

 **return** $\hat{y}_i \quad \forall i$;

---

## A.4    Computing k-shot 5-way accuracy:

We follow the same protocol as in Santoro et al. [13]. k-shot results are calculated based on the prediction accuracy of $k^{th} + 1^{th}$ instance of the class– that is, when the model sees an image of the class for the $k$ times.

## A.5    Architecture Details

The LSTM baseline and the controller in NTM have a hidden size of 400, followed by a 5-way classifier. The non-parametric memory in NTM is a matrix in $\mathbb{R}^{120 \times 40}$, with 0.95 as the decay rate. In order to make comparisons fair, we use a 4-layer Convolutional LSTM with a 64 channels in each layer, followed by a LSTM of hidden size 400 and a 5-way classifier.

For both OPN and CPM, we use the same features extractor as in Ren et al. [18]: a 4-layer CNN with 64 channels in each layer. The CPM has, in addition to the feature extractor, an LSTM of hidden size 400, followed by a gating a mechanism for updating prototypes. We adopt the same feature extractor when training APL, along with a LSTM decoder with hidden size 400 and a total of 5 neighbors read from the memory for each sample. Note: In the APL paper they use a ResNet as the feature extractor.

All models are trained with a learning rate of $10^{-3}$ using Adam [57] for 100k iterations with a meta batch size of 16.

# B    Task Length Dependent Optimization Issues

In initial experiments, we observe that for tasks with 50 time steps these models did not train well. We hypothesize that this could be due to the same network being repeated 50 times, thereby inducing an effective depth of the network to be $6 \times 50$ (since our model has 6 layers). We resolve training difficulties by adding a skip connection, in the style of residual networks [54], between the second layer and the fourth layer (omitted in Figure 2). This makes a significant impact in convergence and generalization.

From Table 6, we see that regardless of actual depth, effective depth seems to be critical in determining if the model can learn an efficient task adaptation strategy. In these experiments, we use a task protocol that is slightly different from the protocol in Section 3.1. Here, we have a separate support and query set, and the model is evaluated only on its performance on the query set. The support set is presented to the model in an online manner, one character at a time step. For models with depth 4, we have the following architecture: 2 ConvLSTM layers followed by a LSTM and a classifier. For models with depth 6, we use 4 ConvLSTM layers followed by a LSTM and a classifier. As mentioned in the paper, we ameliorate training issues by adding a skip connection to these models.

| Network Depth | Task | Task Length | Effective Depth | Efficient Task Adaptation |
|---|---|---|---|---|
| 4 | 5-shot 6-way | 30 | 120 | ✓ |
| 4 | 5-shot 7-way | 35 | 140 | ✗ |
| 6 | 5-shot 4-way | 20 | 120 | ✓ |
| 6 | 5-shot 5-way | 25 | 150 | ✗ |

Table 6: **Effect of task length on learning efficient task adaptation strategies.** Effective depth = network depth $\times$ task length, seems to determine if the network can learn efficient adaptation rules. If effective depth is greater than 140, the model fails to adapt efficiently to a given task.

# C    Details of Continual Learning Experiments

## C.1    Training

We always start by exposing the model to a 5-way continual learning problem, after which for every 5K updates we increase the length of the tasks exposed to the model by 1 till the exposed task length is equal to the desired task length. Example: Suppose 7-way 5-shot is the desired task we want our model to solve, we start with a 5-way task for the first 5K updates, then 6-way for the next 5K

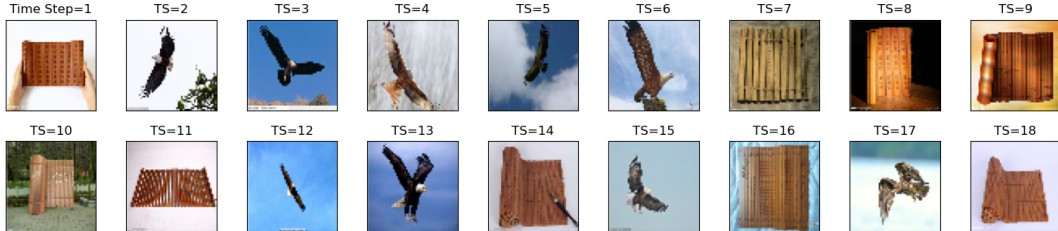

Figure 7: **A sample online segmentation task with distractors:** A sequence of parchment and eagle images are presented to the model (one image at a time step). Here parchments are the distractor images and the eagles are the images to be segmented.

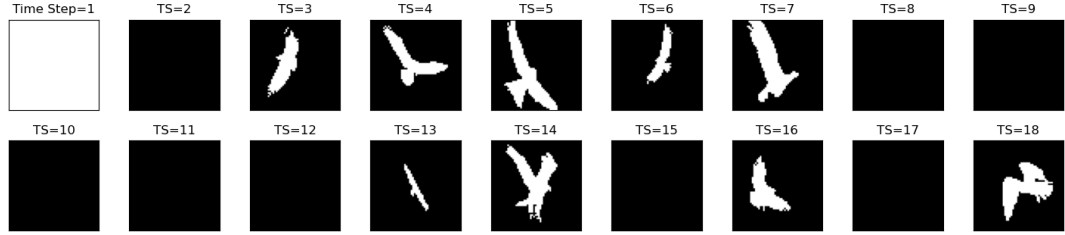

Figure 8: **Ground truth segmentation labels for the above task.** The first time step we present a offset null label as mentioned in the Section 4.4. In the subsequent time steps, we provide the model with ground truth label for the previous time step's input image. The ground truth corresponding to the parchment image is the all zero tensor since it has to be masked out.

updates, and then finally we train the model on 7-way tasks until convergence. We train CL+LSTM with a meta-batch of 16 using Adam and a learning rate of $10^{-3}$.

## C.2 Details of OML Experiments

We use publicly released code[1] with the suggested hyperparameters (Adam/SGD with learning rates $10^{-4}$/0.03 for meta-training/adaptation), while adopting a 4-layer CNN with a parameter count comparable to ours. Unlike Javed and White [6], we use $28 \times 28$ images instead of $84 \times 84$ and train for a total of 150k episodes. We use SGD for adaptation instead of Adam in meta-testing, which resulted in significantly better results for our setting where trajectories are short, likely due to the lack of momentum.

## C.3 Details of ANML Experiments

We use publicly released code[2] with the suggested the hyperparameters: meta-learning rate $10^{-3}$, Adam outer loop optimizer, and gradient descent as the inner loop optimizer. We use a 4-layer CNN with a 256 filters and a 3-layer hypernetwork yielding a model with parameter count 3 times that of our CL+LSTM model. Similar to OML, ANML suffered in generalizing to the last subtask due to momentum issues and using SGD as in the case of OML did not help alleviate the issue. Hence, we used Adam for adaptation as in the original paper.

## D   Segmentation Experiments

### D.1   Dataset

In our segmentation experiments we use the FSS1000 dataset [50], which contains 1000 semantic labels and each label containing 10 images. The dataset is split into 700 training, 60 validation, and 240 test classes. We resize these images into $56 \times 56$ spatial resolution.

---

[1] https://github.com/khurramjaved96/mrcl
[2] https://github.com/uvm-neurobotics-lab/ANML

### D.2 Task Details

A sample online few-shot segmentation task is presented in Figure 7 and its corresponding ground truth is presented in Figure 8. In this task adaptation on the input images is essential as the model has to (on the fly) learn the distractor concept and mask it out, while segmenting the other images.

### D.3 Training Details

Both the CL U-Net models are trained with a learning rate of $10^{-3}$ using Adam optimizer and a batch size of 4 throughout the entire curriculum mentioned in Section 4.4. Further following [50] we train our models with mean squared error loss.

The fine tuning baseline is MAML pre-trained on online few-shot segmentation with no distractors similar to [38]. We train the model with an outer loop learning rate of $10^{-3}$ using Adam and an inner loop learning rate of $10^{-2}$ using gradient descent. The results in Table 4 for the fine tuning baseline is obtained by taking such a MAML pre-trained model and fine tuning it on online few-shot segmentation tasks with distractors. When we tried meta-training this model on few-shot segmentation with distractors, the model converged to a trivial solution of masking out both the distractor images and the images that were to be segmented. Hence we only present the fine tuning results.

### D.4 Model Details

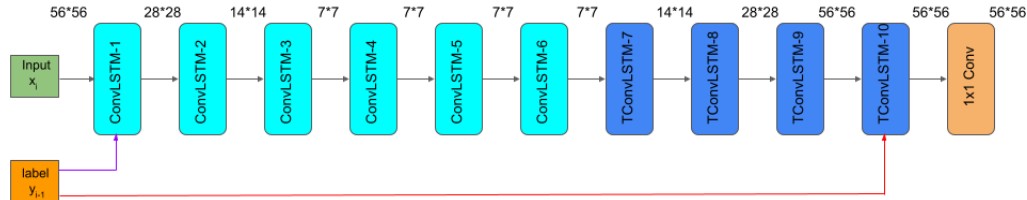

Figure 9: **CL U-Net: Augmenting memory cells to layers of a U-Net variant.** The cyan layers are ConvLSTM layers, and the reduction in spatial resolution is brought about via pooling layers (not in figure). The dark blue layers (TConvLSTM) are transpose convolutional LSTM layers and are used to upsample along the spatial dimension. Finally, a $1 \times 1$ conv layer is used to generate logits at a pixel level. The spatial resolution of the activation at each layer is denoted at the top of each layer. We have two modes of label injection: 1) the ground truth label $y_{i-1}$ corresponding to the image at the previous time step is fed to the first layer (purple line); 2) the ground truth label of previous time step image $y_{i-1}$ is presented to the $10^{th}$ layer (red line). We observe that the first mode (purple) is more effective than the second mode (red). Skip connections are not shown for sake of clarity: skip connections exist between layers 2 and 4; layers 4 and 6; layers 6 and 8.

In Figure 9, we use a variant of U-Net wherein each layer is augmented with memory cells. Unlike U-Nets, we do not have skip connections from the first layer to the last layer; we employ skip connections similar to ResNets–once every two layers between the $2^{nd}$ layer and the $8^{th}$. In U-Nets, it is conceivable that label information propagation is easier even to the last layer of the network, since there is a direct connection between the first and the last layer. However, the effectiveness of label injection in a CL U-Net (as evidenced by Table 4) demonstrates that effective label information propagation can take place in standard networks like ResNets as is, without having to employ any additional pathways or tricks. This underlines that label injection can be used for a vast variety of standard networks.

We replace the ConvLSTM cells with standard CNN layers and use ReLUs as nonlinearities to get the architecture for the CNN based finetuning baseline.

# E    Supervised Learning Experiments

## E.1    Datasets: CIFAR10 (C10) and CIFAR100 (C100)

C10 and C100 consist of 60000 RGB images of size $32 \times 32$ sampled from 10 and 100 classes, respectively. We adopt the standard split where 50000 images are used for training and 10000 for testing. Following He et al. [54], we perform horizontal flips with $50\%$ probability and adopt random translations by padding 4 pixels to each image and then selecting a randomly-chosen $32 \times 32$ crop.

## E.2    Architecture Details

To evaluate how ConvLSTM layers impact the performance of models typically adopted for this task, we use VGG-11 [53] and ResNet-20 [54] as base networks and replace each convolutional layer, along with the following batch norm [58] and ReLU operations, by a ConvLSTM with recurrent batch norm [59]. We refer to the produced models as 'CL-VGG-11' and 'CL-ResNet-20'. Further, we shrink the filter set and it is denoted as the following: CL-VGG-11 $(0.5\times)$ has half as many filters per layer as CL-VGG-11, and produces activation tensors with 32 channels instead of the 64 in VGG-11.

Although ConvLSTM variants differ from their original counterparts (*e.g.,* ReLU is replaced by the tanh activation, which is known yield harder optimization due to saturation), our chosen hyperparameters and training setup were adopted from He et al. [54] and designed for the base networks, hence the performance of CL-VGG-11 and CL-ResNet-20 can likely be further improved by hyperparameter tuning.

## E.3    Training Details

Each network is trained for a total of 200 epochs with SGD and an initial learning rate of $0.1$, which is decayed by a factor of 10 at epochs 100 and 150, following He et al. [54]. We also employ Nesterov momentum of $0.9$ and a weight decay of $10^{-4}$. For the ConvLSTM variants, we only backpropagate errors up to 5 previous mini-batches, akin to truncated backpropagation through time [60] with a time window of 5 steps. Cell states are frozen prior to inference to avoid information leakage across test samples.