# OpenReview forum: "Online Meta-Learning via Learning with Layer-Distributed Memory"
_NeurIPS.cc/2021/Conference — NeurIPS 2021 Poster_

### Official Review · Reviewer_U3Jn · 2021-07-14

**Rating:** 6
**Confidence:** 4

**Summary:**

In this paper the authors propose a distributed memory system to tackle the online few-shot setting. The system is based on layers of conv-LSTMs that take as input an image and the labels at previous steps to generate activations and store information.

Overall I think that the technical contribution is marginal. Both the system structure and the problem formulation make the method impractical and difficult to scale to more complex settings. Experiments are limited to small datasets and do not show the potential of the proposed solution. Limitations are not appropriately discussed and a better framing w.r.t. more recent online few-shot learning work is missing. In the current form the paper is not ready to be accepted. I am open to change my mind and increase my score if the authors provide a strong rebuttal and a satisfactory answer to the concerns I have outlined below.

**Limitations And Societal Impact:**

Yes

**Main Review:**

1) Clarification on the importance of having a memory. I have checked algorithm 1 in the appendix and the details of how task are sampled and aggregated in appendix A.1. The authors wrote: "we sample 10-shot 5-way online few-shot tasks by sampling 5 classes and 10 samples per class (a total of 50 images), which are fed to the model at a rate of one image per time step". My understanding is that those 50 images are shuffled and presented in a random order. I am wondering if there are simpler ways to manage this setting, for instance by just accumulating the classes seen so far into bins and then pass the histogram instead of the one-hot label. This could be done by simply summing the one-hot vectors seen so far (and optionally normalize by the total number of images in the sequence). One could have a standard neural network that makes a prediction by using the image at time t and the accumulated histogram. If I am correct in my assumptions, then this seems to be a trivial baseline to try in order to justify the use of a memory. In other words, why we need a complex memory mechanism if the only information that matters is the current image and the labels accumulated so far in the sequence? The authors should clarify this point and provide empirical evidences that this trivial baseline is not as competitive as the proposed method.

2) Problem setting. The particular formulation used in this work has been used previously, for instance in [1], and it is based on the key assumption that labels are delayed by a given factor. However, this formulation is just one of the possible ways meta-learning can be applied to the online setting, and it may suffer several shortcomings when used in practice. For instance, in many real-world settings the feedback can be delayed by a variable factor, it can change between train and evaluation stages, and even be overwritten by inputs belonging to different classes. Recent work has investigated online few-shot learning in some of those challenging settings but the authors did not mention it or discuss it. I suggest including these papers [2,3,4] with an explanation of the advantages/disadvantages of the formulation used by the authors.

3) Computational cost. A satisfactory discussion of computational costs and memory issues has not been provided. The system seems quite expensive in terms of computational resources. The authors pointed out in lines 154-157 some stability issues that they assume are due to: "the same network being repeated 50 times". If backpropagation is performed from the last step to the first one, it will require the storage of all the activations along the way. This is a serious limitation, the model may not scale well to large images and deeper backbones due to this bottleneck. A discussion of all these issues is expected.

4) Datasets. The datasets used for the evaluation of the proposed method are Omniglot and CIFAR-FS. I think that Omniglot is starting to be outdated and has been replaced by more sophisticated benchmarks in recent publications (e.g. Meta-Dataset [5], SlimImageNet [2]). CIFAR-FS has also some serious limitations (limited diversity, small resolution, etc) when compared to recent benchmarks. The authors are encouraged to move towards more recent benchmarks if they want to fully showcase the potential of the proposed solution.


References
----------

[1] Santoro, A., Bartunov, S., Botvinick, M., Wierstra, D., & Lillicrap, T. (2016). One-shot learning with memory-augmented neural networks. arXiv preprint arXiv:1605.06065.

[2] Antoniou, A., Patacchiola, M., Ochal, M., & Storkey, A. (2020). Defining benchmarks for continual few-shot learning. arXiv preprint arXiv:2004.11967.

[3] Caccia, M., Rodriguez, P., Ostapenko, O., Normandin, F., Lin, M., Caccia, L., ... & Charlin, L. (2020). Online fast adaptation and knowledge accumulation: a new approach to continual learning. arXiv preprint arXiv:2003.05856.

[4] Ren, M., Iuzzolino, M. L., Mozer, M. C., & Zemel, R. S. (2020). Wandering within a world: Online contextualized few-shot learning. arXiv preprint arXiv:2007.04546.

[5] Triantafillou, E., Zhu, T., Dumoulin, V., Lamblin, P., Evci, U., Xu, K., ... & Larochelle, H. (2019). Meta-dataset: A dataset of datasets for learning to learn from few examples. arXiv preprint arXiv:1903.03096.

**Time Spent Reviewing:**

3

---

> ### Author Response · Authors · 2021-08-11
> **Response to Reviewer U3Jn**
>
> # Importance of memory:
> In Table 2, the CNN+LSTM and CNN-F +LSTM models are ablation studies to underscore exactly the importance of memory.  In these experiments, we replace our ConvLSTM architecture with a CNN architecture.  In CNN+LSTM the CNN is episodically trained, and further we consider a variant CNN-F +LSTM, in which we take a pre-trained CNN and freeze its weights (no episodic training). As in Table 2, we see that the CL+LSTM model outperforms memoryless baselines.  This strongly suggests that memory is essential for adapting the feature extractor at different time steps.
>
> The assumption that, "the only information that matters is the current image and the labels accumulated so far in the sequence" is incorrect.  The information in all images (e.g., features built from their contents) seen so far in the sequence matters -- one wants to learn and refine a model of each of the classes as more examples are obtained.  Using a simple hand-designed histogram is an impoverished representation compared to allowing the system to learn relevant features to store in a layer-distributed memory. Further in our online tasks such a model will fail to make any associations between samples and class concepts and will lead to a random classifier's performance.
>
> # Problem setting:
> The online problem setting followed by [1] assumes that the label for a given sample is revealed after the prediction is made, which is a standard assumption made in several works [2].  Our focus is primarily on this popular setting.  We agree that the quantum of delay can vary, and hence we went beyond the standard setting and conducted the experiments in Table 3, where we vary the delays on the label. As mentioned in **line 228**, memory is essential for the network to be able learn the pattern in label delay.  For prototypical and gradient-based methods, the delay would have to be known a priori for any learning to happen; this is not required for our CL+LSTM model.
>
> # Related problem settings:
> 1) Antoniou et al. [2] present benchmarks for continual few-shot learning.  The network is presented a number of few-shot tasks, one after the other, and then is expected to generalize even to the previously seen tasks.  This is a challenging and interesting setup, in that, the network has to show robustness to catastrophic forgetting while learning from limited data. However, the ability to master a subtask when samples are provided in an online manner is not tested in this framework.  In lines 51-54, we discuss the challenges involved in training when samples for a subtask are provided in an online manner. Moreover, the maximum number of subtasks they experiment with is 10, and since all the data pertaining to a subtask is presented at the same time step, the model at most has to remember inductive biases acquired only 9 time steps ago. In our online continual learning benchmark, wherein we follow established baselines such as OML and ANML, the model has to remember inductive biases acquired 90 to 100 times steps ago, thereby serving as a robust measure for immunity against catastrophic forgetting.
>
> 2) Caccia et al. [3] present benchmarks that measure the ability of a model to adapt to a new task using the inductive biases that it has acquired over solving previously seen tasks.  More specifically, the benchmark presents an online non-stationary stream of tasks, and the model's ability to adapt to a new task at each time step is evaluated.  Note that they do not measure the model's ability to remember earlier tasks; they only want the model to adapt well on a newly presented task.  This is a departure from our setup, wherein we care about the model's ability to remember tasks seen several time steps ago.
>
> 3) Ren et al. [4] discuss an online few-shot learning environment with a certain structure where images are sampled with a certain spatio-temporal proximity.  This structure is supposed to capture visual experience of a wandering agent.  While this environment is indeed interesting, we explore an unstructured online stream of data, where learning the class concepts in a task is paramount in making future predictions, and the model cannot rely on spatio-temporal cues for learning class concepts.
>
> Thank you for asking us to draw these distinctions, we find this discussion crucial and will include this in the next revision.
>
> # Computational cost:
> First, we want to underscore that during inference time, adaptation via hidden states is cheaper than computing gradients for a similarly sized network.  During inference, our memory network need only operate in forward mode, whereas for gradient-based methods both a forward pass and a backward pass are required.
>
> Second, these stability issues are seen across different meta-learning approaches owing to the task specific adaptation part of the episodic training paradigm. MAML suffers from similar stability issues since backpropagating through $n$ steps of SGD is similar to backproping through a network that is repeated $n$ times [5]. Similar to us, they add skip connections to ameliorate this issue.
>
>
> In our online semantic segmentation experiments, we train a 10-layer deep network on a single 2080-Ti GPU. However, to ease  concerns on the scalability of our model, we have run new semantic segmentation experiments on a 16-layer UNet like backbone: our results show that despite scaling to a deeper network, no optimization snags were encountered.  See below table for mIoU values.
>
> | Model  | Image Resolution    |Label Injection     |Segmentation           | Masking                 |
> |-----------|---------------------------|------------------------|----------------------------|---------------------------|
> |             |                                 |                            | 1-shot      ,      6-shot | 1-shot  ,  6-shot       |
> CL-UNet-16|   112 x 112        |      1st layer         | 52.23         ,   64.29 | 89.4,   92.75  |
>
>
> # Datasets (also covered in reply to Reviewer s4fj):
> The online setting in which we experiment is more difficult than the standard few-shot classification setting.  In the online setting, smaller datasets still present a significant challenge.  In the standard few-shot learning scenario, wherein the entire support set (all training samples) is available at the same time step, several models have achieved high accuracies 5-shot of 99.9\% and 86.9\% on Omniglot and CIFAR-FS [6,7]; consequently, the community has moved to harder datasets like miniImageNet and tieredImageNet.  However, in the online setting, even the more recent methods (Ren et al.) obtain only around 97\% and 55.8\% on 5-shot online Omniglot and CIFAR-FS, as shown in Table 1.  Lines 52-54 (Section 1) outline the additional technical challenges associated with the online setting.  We believe it is crucial that this gap between the online and standard settings is settled for Omniglot and CIFAR-FS.  Furthermore, for the online continual learning experiments, both OML and ANML are established baselines and their main results only use Omniglot.
>
> On the subject of scaling with image resolution, the semantic segmentation tasks are on the FSS1000 dataset, where the image resolution is $56 \times 56$.  To demonstrate further scalability, we increase the resolution of our segmentation experiments to $112 \times 112$.  In the below table, we see that the memory-based CL-UNet model improves with increased resolution, strongly suggesting that the adaptation rule learned by the network scales well with resolution, even for difficult tasks like semantic segmentation.
>
> | Model  | Image Resolution    |Label Injection     |Segmentation           | Masking                 |
> |-----------|---------------------------|------------------------|----------------------------|---------------------------|
> |             |                                 |                            | 1-shot      ,      6-shot | 1-shot  ,  6-shot       |
> CL-UNet|   56 x  56                 |      1st layer         | 49.2         ,     56.0    | 84.6      ,   91.7        |
> CL-UNet|   112 x 112              |      1st layer         | 49.5         ,   **64.1** | **88.9** ,   **94.3**  |
>
>
> # References
> [1] Santoro, A., Bartunov, S., Botvinick, M., Wierstra, D., & Lillicrap, T. (2016). One-shot learning with memory-augmented neural networks. arXiv preprint arXiv:1605.06065.
>
> [2] Antoniou, A., Patacchiola, M., Ochal, M., & Storkey, A. (2020). Defining benchmarks for continual few-shot learning. arXiv preprint arXiv:2004.11967.
>
> [3] Caccia, M., Rodriguez, P., Ostapenko, O., Normandin, F., Lin, M., Caccia, L., ... & Charlin, L. (2020). Online fast adaptation and knowledge accumulation: a new approach to continual learning. arXiv preprint arXiv:2003.05856.
>
> [4] Ren, M., Iuzzolino, M. L., Mozer, M. C., & Zemel, R. S. (2020). Wandering within a world: Online contextualized few-shot learning. arXiv preprint arXiv:2007.04546.
>
> [5]  Antoniou, Antreas, Harrison Edwards, and Amos Storkey. "How to train your MAML." arXiv preprint arXiv:1810.09502 (2018).
>
> [6]  Finn, Chelsea, Pieter Abbeel, and Sergey Levine. "Model-agnostic meta-learning for fast adaptation of deep networks." International Conference on Machine Learning. PMLR, 2017.
>
> [7] Tian, Yonglong, et al. "Rethinking few-shot image classification: a good embedding is all you need?." Computer Vision–ECCV 2020: 16th European Conference, Glasgow, UK, August 23–28, 2020, Proceedings, Part XIV 16. Springer International Publishing, 2020.

---

> > ### Comment · Reviewer_U3Jn · 2021-08-14
> > **Response to authros**
> >
> > I would like to thank the authors for their answer that has clarified some of my doubts. I have read both the rebuttal and the comments of the other reviewers and I still have some concerns that I would like to detail below.
> >
> > 1) Problem setting. I share the same doubts of Reviewer s4fJ about the clarity of the paper and the problem setting. This confusion may be due to the number of ways few-shot and continual learning have been joined together. The authors work is based on the setup described in [4], which is a paper from 2016. In the meanwhile the community has moved towards other settings (e.g. [1,2,3]), some of them more challenging. While the authors have provided an answer in the rebuttal by comparing with [1,2,3], what is still missing is a clear motivation for the proposed setting. I would like to have clear answers for the following questions: What makes the current setting important? Which kind of problem is the current setting trying to solve? Are there any real-world applications that could be formalized by using this setting?
> >
> > 2) Datasets. The authors have justified the use of Omniglot and CIFAR-FS saying that they want to settle the gap between the online and the standard setting for those datasets before moving to something more complex. In my opinion the use of more complex datasets is not just matter of accuracy quantification, it gives important information about scalability and performance degradation as the difficulty of the task increases. Note that, the authors cite Ren et al. (2020) to justify the performance gap in Omniglot and CIFAR-FS, but Ren et al. (2020) have also provided results on a more challenging dataset (RoamingRooms). Similar papers [2,3] also report the performance on variants of ImageNet showing how performances and memory usage vary across datasets. This further support my point.
> >
> > 3) Computational cost. This has not been answered in a satisfactory way. The authors have provided additional experiments on the FSS1000 dataset using $112 \times 112$ images. However, the details of these experiments are scarce. How long is the sequence? Which kind of model has been used? I understand that a robust set of experiments was hard to provide for the rebuttal, but the results provided so far are not enough. Here I would like to clarify why I think memory analysis is important in this context. The memory issues in continual few-shot learning are due to two factors: (i) image size, and (ii) sequence length (see [2] for a detailed discussion). It is unclear how the method trade-off between the two. To settle this issue it would be sufficient to report the performances in terms of MACs and wall-clock time for sequences of varying length and different image resolution. Ideally this should be done for both training and evaluation. I am not sure if this could be done during the discussion phase, but in my opinion it would improve the quality and impact of the paper.
> >
> >
> > [1] Ren, M., Iuzzolino, M. L., Mozer, M. C., & Zemel, R. S. (2020). Wandering within a world: Online contextualized few-shot learning. arXiv preprint arXiv:2007.04546.
> >
> > [2] Antoniou, A., Patacchiola, M., Ochal, M., & Storkey, A. (2020). Defining benchmarks for continual few-shot learning. arXiv preprint arXiv:2004.11967.
> >
> > [3] Caccia, M., Rodriguez, P., Ostapenko, O., Normandin, F., Lin, M., Caccia, L., ... & Charlin, L. (2020). Online fast adaptation and knowledge accumulation: a new approach to continual learning. arXiv preprint arXiv:2003.05856.
> >
> > [4] Santoro, A., Bartunov, S., Botvinick, M., Wierstra, D., & Lillicrap, T. (2016). One-shot learning with memory-augmented neural networks. arXiv preprint arXiv:1605.06065.

---

> > > ### Author Response · Authors · 2021-08-20
> > > **Addressing Concerns on Problem setting, Datasets, and Computational Cost**
> > >
> > > We thank the reviewer for taking time to go over individual responses.
> > >
> > > 1) **Problem setting:** In our online continual learning  experiments (Sec:4.3), we follow OML (2019) [5] and ANML (2020) [6]; ANML and OML are widely cited baselines (roughly 150 citations), with recent focus from the community. Furthermore, there is value in showing that recent online methods such as APL, OPN, CPM as implemented in [7,1], are not close to saturation (Table 1 in our paper), even on older and simpler benchmarks as proposed by *Santoro et al.* [4] (2016). It is interesting that a ConvLSTM model such as ours can significantly outperform these recent online baselines (Table 1, paper).
> > >
> > >     **Significance of our setting:** In part, the motivation for the current setting is outlined in our comparisons with [1,2,3] (earlier response): our online few-shot experiments (Section 4.1) capture a model's ability to learn a class concept from an online stream of only samples and labels (unlike Ren et al., where models can rely on spatio-temporal cues); Our continual learning experiments measure a model's ability to remember tasks seen earlier (say 100 time steps ago) in a given online stream (*Caccia et al.* don't measure ability to remember; *Antoniou et al.* operate in a batch setting, plus the task requires models to remember only across a few time steps).
> > >
> > >     **What makes the current setting important?** Our setting measures the above mentioned aspects of a model's capabilities that are not measured by [1,2,3].  There is an argument for the importance of the online versions of few-shot (Section 4.1) and continual learning scenarios (Section:4.3): rapid task adaptivity is likely to be valuable in dynamic real-time environments. Substantial prior effort has been devoted solely to the online setting [1,3].
> > >
> > >     **Are there any real-world applications that could be formalized by using this setting?** Suppose we are tasked with presenting a user with recommendations (for some entity), and the user can like or dislike each recommendation.  The user's feedback (likes/dislikes) creates a stream of labelled data, on which our model could adapt with just few samples, and learn to provide a personalized set of recommendations.  This is similar to the online few-shot learning experiments in Table 1 (paper).
> > >
> > >     Consider a similar setup where the user's behaviour changes temporally (perhaps, looking for entity $x$, then searches for entity $y$, and then goes back to $x$); this is similar to the continual learning setup (Figures 3, 4; paper).  Here, we have a non-stationary online stream, on which we want to adapt.
> > >
> > > 2) **Datasets:** The gap between online and standard settings is not just as a matter of accuracy quantification.  There is a qualitative point here: the significant gap implies that the current online methods are failing to learn crucial features for simpler datasets -- ones that methods in standard settings seem to learn easily, thereby suggesting fundamental limitations in the nature of adaptation schemes employed by current online models.  We would like to overcome these limitations in online learning.  We do agree with the reviewer in that information about scalability and performance degradation is obtained as the difficulty of the task increases.  We have online few-shot segmentation experiments (Section 4.4), wherein we increase the difficulty of the task (from classification to segmentation) and use a more complex and recent dataset FSS-1000 [8].  In Table 4 (paper), we demonstrate that our memory-based model outperforms the fine-tuning baseline --which is an adaptation of MAML -- for the task of online few-shot segmentation.  This result suggests that adaptation via memory is more suitable for a challenging task like segmentation.
> > >
> > > 3) **Computational Cost:** We agree that sequence length in continual learning tasks is a crucial parameter in determining memory load.  The long adaptation phase in the training episodes of our experimental environment adds to the memory load.  However, this memory load is not unique to our method.  In fact, if adaptation is desired at every time step of the task, then other meta-learning methods will be at least be as costly as ours.  Meta-learning algorithms such as MAML (and variants), which seek to adapt during each time step, also incur roughly the same memory load during training as our model; MAML models require backproping through $t$ (= sequence length) steps of gradient descent [9].  Since these models have to store the computational graph for each time step, they are required to store activations at each time step for each layer.  As an additional consequence, they also scale similarly with image resolution.
> > >
> > >     Post-training, when the model is presented with an unseen task at inference time, our model can adapt with lesser computational load compared to gradient-based models -- assuming adaptation is required at every time step. To support this claim, we present experiments in the below Table.  The OML baseline uses the most GFLOPs; indeed, it is a deeper network: 6 conv and 2 fully connected layers; Our CL+LSTM models have fewer parameters and consequently fewer GFLOPs. The C4 model has the same depth and roughly the same parameter size our CL+LSTM model and uses fewer GFLOPs per forward pass; this is due to the absence of additional gating operations that the CL+LSTM model has to perform.  However, at inference time when the model has to adapt to a new task, the C4 model consumes more GFLOPs than our CL+LSTM.
> > >
> > >     | Model  | Num. Parameters    | GFLOPs per forward pass     | 5 Sub-Task Continual Learning Accuracy|
> > >     |-----------|---------------------------|----------------------------------------|-------------------------------------------------------|
> > >           C4   |   1.88 M                   |      0.30                                   |                          NA                                   |
> > >     OML      |   5.32 M                        |      1.46                                   |                    94.60                                     |
> > >     CL+LSTM | 1.81 M                 |     0.40                                     |                          98.31                               |
> > >
> > >     Note that we use the following computation to determine the GFLOPs consumed while adapting to an unseen task: (GFLOPs per forward pass) * (3 for forward and backward pass) * (task length); note that this is a minor modification to method 1 as adopted in this [OpenAI blog](https://openai.com/blog/ai-and-compute/).  The factor of 3 for forward and backward pass is because the backward pass is assumed to be twice as costly as the forward pass (since it computes gradients and performs updates).  So the total GFLOPs per task for the C4 model = 0.30 * 3 * $t$ = 0.9 $t$ GFLOPs; where $t$ is task/sequence length. Since we do not have any backprop operation for task adaptation and simply run our model in forward mode, we drop the factor of 3. Hence, the GFLOPs consumed by CL+LSTM model = 0.40 * 1 * $t$ = 0.40 $t$.  Notice that the GFLOPs for both CL+LSTM and C4 model increases linearly with task/sequence length $t$.  This factor of $t$ is a direct consequence of seeking to adapt at each time step; this is crucial in dynamic online learning settings.  In terms of time taken to complete one time step, our CL+LSTM model takes 5.76 (ms), while OML model takes 10.33 (ms); in part, the difference here is because our model runs in forward mode at inference time, while OML requires both forward and backward passes.
> > >
> > >     Most of our analysis here is devoted to measuring performance in inference time, and we observe a clear edge in FLOPs and wall clock time for our CL+LSTM model because of its ability to any avoid gradient computation at inference time.  During training time, we lose this advantage since we perform BPTT to update our model.  We suspect BPTT and computing meta-gradients (as in our baselines) should be comparable in compute cost, as both involve backpropagating through instances of the same network multiple times.  We will add experiments in the revised paper to quantify this comparison.
> > >
> > > **References:**
> > >
> > > [1] Ren, M., Iuzzolino, M. L., Mozer, M. C., \& Zemel, R. S. (2020). Wandering within a world: Online contextualized few-shot learning. arXiv preprint arXiv:2007.04546.
> > >
> > > [2] Antoniou, A., Patacchiola, M., Ochal, M., \& Storkey, A. (2020). Defining benchmarks for continual few-shot learning. arXiv preprint arXiv:2004.11967.
> > >
> > > [3] Caccia, M., Rodriguez, P., Ostapenko, O., Normandin, F., Lin, M., Caccia, L., ... \& Charlin, L. (2020). Online fast adaptation and knowledge accumulation: a new approach to continual learning. arXiv preprint arXiv:2003.05856.
> > >
> > > [4] Santoro, A., Bartunov, S., Botvinick, M., Wierstra, D., \& Lillicrap, T. (2016). One-shot learning with memory-augmented neural networks. arXiv preprint arXiv:1605.06065
> > >
> > > [5] Javed, Khurram, and Martha White. "Meta-learning representations for continual learning." arXiv preprint arXiv:1905.12588 (2019).
> > >
> > > [6] Beaulieu, Shawn, et al. "Learning to continually learn." arXiv preprint arXiv:2002.09571 (2020).
> > >
> > > [7] Ramalho, Tiago, and Marta Garnelo. "Adaptive posterior learning: few-shot learning with a surprise-based memory module." arXiv preprint arXiv:1902.02527 (2019).
> > >
> > > [8] Li, Xiang, et al. "Fss-1000: A 1000-class dataset for few-shot segmentation." Proceedings of the IEEE/CVF Conference on Computer Vision and Pattern Recognition. 2020.
> > >
> > > [9] Antoniou, Antreas, Harrison Edwards, and Amos Storkey. "How to train your MAML." arXiv preprint arXiv:1810.09502 (2018).

---

> > > > ### Comment · Reviewer_U3Jn · 2021-08-21
> > > > **Final response to authors**
> > > >
> > > > Thank you for the detailed answer. I will increase my score to 6. I invite the authors to discuss the following points if the paper gets accepted:
> > > >
> > > > 1) Add a detailed discussion regarding the differences with respect to [1, 2, 3] and the work on distributed memory suggested by Reviewer oAcd.
> > > >
> > > > 2) Include the considerations on computational cost (possibly in a separate subsection) and experiments showing the performance of the method in terms of FLOPs/MACs and wall-clock time against baselines at both training and test time.
> > > >
> > > >
> > > > References
> > > >
> > > > [1] Ren, M., Iuzzolino, M. L., Mozer, M. C., & Zemel, R. S. (2020). Wandering within a world: Online contextualized few-shot learning. arXiv preprint arXiv:2007.04546.
> > > >
> > > > [2] Antoniou, A., Patacchiola, M., Ochal, M., & Storkey, A. (2020). Defining benchmarks for continual few-shot learning. arXiv preprint arXiv:2004.11967.
> > > >
> > > > [3] Caccia, M., Rodriguez, P., Ostapenko, O., Normandin, F., Lin, M., Caccia, L., ... & Charlin, L. (2020). Online fast adaptation and knowledge accumulation: a new approach to continual learning. arXiv preprint arXiv:2003.05856.

---

### Official Review · Reviewer_oAcd · 2021-07-16

**Rating:** 7
**Confidence:** 5

**Summary:**

The paper proposes to adapt Meta RNNs (memory-based meta-learning) by instantiating multiple layers of LSTMs where some of them are convolutional.
The authors refer to this as distributed memory due to each layer having its own LSTM-based memory.
They demonstrate good performance in few-shot learning and continual learning.

**Limitations And Societal Impact:**

Limitations and potential negative societal impact is adequately discussed.

**Main Review:**

## Originality and Significance

The authors suggest that instead of using a single LSTM in MetaRNNs (memory-based meta-learning) multiple layers should be added and some should be convolutional.
While this seems like a fairly incremental change, they demonstrate convincingly that this helps in few-shot learning and continual learning.
In particular, the improvement / similar performance over ANML and OML is interesting as no inner gradients are required.

## MetaRNNs (memory-based meta-learning) not correctly cited

The proposed architecture is very similar to Meta RNNs (supervised [1], RL [2,3]) that the authors do not cite. They only discuss other forms of external memory in the related work section. The author's contribution of adding multiple layers of LSTMs and making some of them convolutional should be highlighted more clearly. This contribution tends to get lost in the paper over discussions that also apply to previous work.

[1] Hochreiter, S., Younger, A. S., & Conwell, P. R. (2001). Learning to learn using gradient descent. International Conference on Artificial Neural Networks.

[2] Duan, Y., Schulman, J., Chen, X., Bartlett, P. L., Sutskever, I., & Abbeel, P. (2016). RL^2: Fast Reinforcement Learning via Slow Reinforcement Learning. ArXiv Preprint ArXiv:1611.02779.

[3] Wang, J. X., Kurth-Nelson, Z., Tirumala, D., Soyer, H., Leibo, J. Z., Munos, R., Blundell, C., Kumaran, D., & Botvinick, M. (2016). Learning to reinforcement learn. ArXiv Preprint ArXiv:1611.05763.

## Connections to other recent work on distributed memory / fast weights should be added

Recent work [4] also introduced a form of distributed memory for meta-learning online adaptation strategies / learning algorithms. In that case, each weight in a neural network is replaced by an LSTM, which is a different but related architectural choice compared to the proposed work. The connection should be discussed and cited.

[4] Kirsch, L., & Schmidhuber, J. (2020). Meta Learning Backpropagation And Improving It. ArXiv Preprint ArXiv:2012.14905.

## Conclusion

Overall I think it is a good paper with interesting results on meta-learning with distributed memory. Unfortunately, it is missing some crucial relationships to previous work. I would like to give the paper a good score based on appropriate adjustments in the rebuttal.

**Time Spent Reviewing:**

5

---

> ### Author Response · Authors · 2021-08-11
> **Response to Reviewer oAcd**
>
> **Connections to *Hochreiter, S. et al* [1]:**
> Indeed, similar to us, Hochreiter et al. rely on the memory network to learn its own adaptation rule via just its recurrent states. However, as mentioned, we use multiple layers of ConvLSTMs in order to both efficiently handle image inputs and facilitate adaptivity.  We will add citation to Hochreiter et al. and a more detailed discussion of similarities and differences.
>
> **Connections to *Duan, Y. et al*; *Wang, J. et al* [2,3]**: Both works extend the use of Meta RNNs to RL tasks. They rely on the LSTM weights to learn to generate hidden states that help adaptation for the RL tasks.  While they focus on RL tasks, we extend memory models to efficiently handle challenging vision tasks, like few-shot online semantic segmentation.  We appreciate the pointers and will cite and discuss this related work in the RL domain.
>
> **Connections to *Kirsch, L., & Schmidhuber, J.*  [4]:**
> Kirsch and Schmidhuber introduce an interesting form of weight sharing wherein LSTM cells (with tied weights) are distributed throughout the width and depth of the network, however each position has its own hidden state.  Further, they have backward connections from the later layers to the earlier layers, enabling the network to implement its own learning algorithm or clone a human-designed learning algorithm such as backprop.  Both our model and theirs implement an adaptation strategy purely using just the recurrent states.  The difference, however, is in the nature of the adaptation strategy implemented in the recurrent states.  Similar to conventional learning algorithms, their backward connections help propagate error from the last layer to the earlier layers.  In our architecture, the feedback signal is presented as another input, propagated from the first layer to the last layer.  We will add a more extensive discussion and comparison of architecture design and meta-learning strategies; thank you for the reference.

---

> > ### Comment · Reviewer_oAcd · 2021-08-31
> > **Response**
> >
> > I am happy with the response and ask the authors to revise and contextualize their paper based on the feedback of all reviewers, in particular regarding contribution, description of their approach, and missing essential references such as [1,2,3].
> > I have adjusted my score.
> >
> > [1] Hochreiter, S., Younger, A. S., & Conwell, P. R. (2001). Learning to learn using gradient descent. International Conference on Artificial Neural Networks.
> >
> > [2] Guez, A., Mirza, M., Gregor, K., Kabra, R., Racanière, S., Weber, T., … Lillicrap, T. (2019). An investigation of model-free planning.
> >
> > [3] Kirsch, L., & Schmidhuber, J. (2020). Meta Learning Backpropagation And Improving It. ArXiv Preprint ArXiv:2012.14905.

---

### Official Review · Reviewer_s4fJ · 2021-07-16

**Rating:** 6
**Confidence:** 4

**Summary:**

In this paper, the authors propose a novel network architecture for online few-shot learning. They propose to use an LTSM framework and store the previous information in the hidden states of different layers. They also evaluate their methods on other tasks, such as online continual learning, online few-shot semantic segmentation, and standard supervised learning.

**Limitations And Societal Impact:**

The authors have included the limitations and potential negative societal impact.

**Main Review:**

- ***This paper is poorly organized and very hard to read.*** I have been working on few-shot learning for several years and have read lots of papers in this field. However, after reading the abstract and introduction, I totally don’t understand what they want to do. For example, many concepts occur before they are defined. In Line 2, the authors mention “the persistent state”. What is it? It is not defined. In Line 134, the authors also use “h”. I cannot find the definition for h before this paragraph. Is that a value, a vector, or a matrix?

- ***The motivation for using memory-based methods is unclear.*** I think the authors try to motivate their method in Lines 30-36. However, I have lots of questions about this motivation, e.g., (1) what are the definitions of generality and flexibility in few-shot learning, and (2) why using a straightforward loss formulation and standard optimization techniques is better? After reading the introduction, I still don’t know what is the main challenge in this paper and how the authors address it.

- ***The claim “their method outperforms gradient-based, prototype-based, and other memory-based meta-learning strategies” seems incorrect.*** Most of the meta-learning methods, such as MAML and ProtoNets, evaluate the methods in the standard few-shot classification setting. However, the authors evaluate their methods in the online few-shot learning setting. Besides, the comparing methods (LSTM, NTM, APL, OPN, and CPM) don’t contain a gradient-based method.

- ***The experiments are conducted on very small datasets.*** The authors use CIFAR-FS and Omniglot, which are two small datasets. In standard few-shot classification, we usually use miniImageNet and tieredImageNet. In [18], they use RoamingImageNet, which is based on tieredImageNet. So it is not reasonable to run experiments on such small datasets.

- ***The method section is too simple.*** The authors only use one page for the method section. Many details about the implementation are unclear.


**Time Spent Reviewing:**

3

---

> ### Author Response · Authors · 2021-08-11
> **Response to Reviewer s4fJ**
>
> # Paper organization
> By persistent state, we mean the hidden and cell states of the LSTM cell.  We use $h$, which is commonly used to denote the hidden state vector.  In our next revision, we will add these clarifications.
>
> # Motivation:
>
> 1) **Why use memory?** As mentioned in **lines 23-29**, a vast majority of meta-learning solutions use hand-crafted adaptation rules, e.g., computing prototypes and then performing nearest neighbor, or using gradient descent (MAML and variants).  Whereas, a memory-based system implicitly learns the adaptation rule, as a function of hidden states generated by the network (Figure 1 and lines 30-36).  By experimenting with a memory-based method, we are seeing if the network can discover an adaptation rule that is better than human-designed adaptation rules.
>
>
> 2) **Definitions of generality and flexibility in few-shot learning?** Specifically, in **line 31**, we use generality and flexibility as attributes of memory-based methods and not as attributes of any particular task such as few-shot learning.  It is mentioned  subsequently in **line 32**, that since memory models can handle a variety of meta-learning problems by recasting them as sequence learning tasks, the model is general and flexible enough to be applied to a variety of meta-learning problems (as confirmed in our results).  We emphasize that this level of generality is not the case with most other forms of meta-learning approaches.  For example, while prototypical networks are used in few-shot classification and few-shot segmentation, their implementation differs significantly from application to application.  Likewise, while gradient-based methods have notable successes in few-shot learning, such methods suffer from catastrophic forgetting (as evidenced by our experiments in Figure 3 and 4).  However, as stated in **lines 63-67**, we apply our memory-based method without any changes to a wide variety of problems, and still outperform methods that are hand-crafted for specific tasks.
>
>
> 3) **Advantages of a straightforward loss formulation and standard optimization techniques?**
> As stated in **lines 32-36**, standard training places the burden of adaptation on the network's persistent memory (hidden states).  This means that the network must generate hidden states that are useful for adaptation (**lines 42-43**), thereby effectively training the model to learn an adaptation rule by itself.  This is consistent with the general philosophy, prevalent throughout deep learning, that learning features/rules is often better than hand-crafted counterparts. Further, this is evidenced in our experimental results.  For example, in Figure 4: the adaptation rule learnt by the CL+LSTM model does not suffer from forgetting; in contrast, the human-designed gradient descent adaptation rules (e.g., as in OML, ANML) exhibit such forgetting.
>
>
> # Outperforming gradient-based methods:
> In our online continual learning experiments (Figures 3, 4) we compare with OML and ANML, which are both gradient-based methods.  In our online segmentation experiments, the fine-tuned CNN baseline (Table 4) is also a gradient-based method (essentially a variant of MAML).  Thus, we significantly outperform gradient-based baselines on two different tasks.
>
> # Datasets:
> The online setting in which we experiment is more difficult than the standard few-shot classification setting.  In the online setting, smaller datasets still present a significant challenge.  In the standard few-shot learning scenario, wherein the entire support set (all training samples) is available at the same time step, several models have achieved high accuracies 5-shot of 99.9\% and 86.9\% on Omniglot and CIFAR-FS [1,2]; consequently, the community has moved to harder datasets like miniImageNet and tieredImageNet.  However, in the online setting, even the more recent methods (Ren et al.) obtain only around 97\% and 55.8\% on 5-shot online Omniglot and CIFAR-FS, as shown in Table 1.  Lines 52-54 (Section 1) outline the additional technical challenges associated with the online setting.  We believe it is crucial that this gap between the online and standard settings is settled for Omniglot and CIFAR-FS.  Furthermore, for the online continual learning experiments, both OML and ANML are established baselines and their main results only use Omniglot.
>
> On the subject of scaling with image resolution, the semantic segmentation tasks are on the FSS1000 dataset, where the image resolution is $56 \times 56$.  To demonstrate further scalability, we increase the resolution of our segmentation experiments to $112 \times 112$.  In the below table, we see that the memory-based CL-UNet model improves with increased resolution, strongly suggesting that the adaptation rule learned by the network scales well with resolution, even for difficult tasks like semantic segmentation.
>
> | Model  | Image Resolution    |Label Injection     |Segmentation           | Masking                 |
> |-----------|---------------------------|------------------------|----------------------------|---------------------------|
> |             |                                 |                            | 1-shot      ,      6-shot | 1-shot  ,  6-shot       |
> CL-UNet|   56 x  56                 |      1st layer         | 49.2         ,     56.0    | 84.6      ,   91.7        |
> CL-UNet|   112 x 112              |      1st layer         | 49.5         ,   **64.1** | **88.9** ,   **94.3**  |
>
> # References
> [1]  Finn, Chelsea, Pieter Abbeel, and Sergey Levine. "Model-agnostic meta-learning for fast adaptation of deep networks." International Conference on Machine Learning. PMLR, 2017.
>
> [2] Tian, Yonglong, et al. "Rethinking few-shot image classification: a good embedding is all you need?." Computer Vision–ECCV 2020: 16th European Conference, Glasgow, UK, August 23–28, 2020, Proceedings, Part XIV 16. Springer International Publishing, 2020.

---

> > ### Comment · Reviewer_s4fJ · 2021-08-22
> > **Final comments**
> >
> > The author's feedback partially addresses my concerns.
> > As all other reviewers are positive towards this paper, it is okay for me to accept it.
> > I hope the authors could improve the introduction section and merge the additional explanations in the rebuttal to the final version.

---

### Official Review · Reviewer_9VUm · 2021-08-01

**Rating:** 6
**Confidence:** 4

**Summary:**

The authors approach the solution to the problem of meta-learning as simply running a recurrent network through various tasks and back-propagating to train the RNN to adapt to new tasks (as in some previous works). The main advance of this work is proposing an architecture - a Deep Convolutional LSTM (with few architectural details) and showing that it works very well. The simplicity of this advance is a plus. The authors should change initial writing and clarify what they mean by distributed memory (see below in more detail) but otherwise the paper is easy to read. A good number of experiments is conducted, though it would be good if it was tested in harder tasks.

**Ethical Concerns:**

No.

**Limitations And Societal Impact:**

Yes.

**Main Review:**

There are the basic cons of this work: The first one should be easy to fix: The description from the start with very mysterious - talking about distributed memory, makes a reader wonder what kind of new paradigm have they devised, just to later find out it is just a different RNN architecture. Taking about this as “memory” is misleading because while the RNN activations do store information, often times one considers weights of RNN to be the memory, and learning being the adaptation of the weights - which is not what they are meta-learning (the weights update has a fixed algorithm - back-propagation). The authors should make it clear from the beginning what they mean by memory (=activations of an RNN) and what they mean by distributed (=across layers in deep RNN). Once the reader knows that, the paper is well written and easy to read.

The second issue is that the tasks they use are have quite a short time spans - I am reluctant to call this meta-learning. Nevertheless, these types of problems have been studied before, and this paper makes a nice contribution with a number of experiments. However it is not clear how well would this learning to adapt the activations work in problems with much larger time scales.

Details:

- I would say in the abstract what the basic idea is: e.g. something like “We device a variant of deep convolutional LSTM architecture, the feature of which is a large capacity state of activations, and use standard end-to-end training by back-propagating through sequence of tasks, …”

I would even ideally change a title - distributed memory can mean many things, like adapting weights of a networks and neither I feel like this should even be referred to as meta-learning (due to shot time scale nature of the problems).

- This kind of architecture is indeed quite capable - very similar one was used in RL: “An Investigation of model-free planning”, Guez at al., - you could add the reference.

- 134-142 - May be add here some of you previous relevant references of works that do this (train RNN end-to-end) - end of the second last sentence.

- 202: You mention pre-training but what it is is only mentioned in the sentence after the next one.

- 243: In 240 you said that task has 5 samples, but here you are changing to 5-20 - does it have more examples or the same 5 are repeated more. If the former, shouldn’t the performance of the model, Figure 3, theoretically increase since you see more examples of a class?

- 247: Meaning that every query example is started with the same state? (That after training set of tasks)

- Figure 4: I don’t understand what is on the x axis. I thought we have a sequence of tasks, and then test the model on the query set containing samples from all the subtasks.

- Section 4.5: How do you do the training here? Do you sample examples from training set at random and treat them as a long sequence, back-propagating every n steps, or how is this done? What is the architecture of CL-ResNet-20 say? Resnet followed by convLSTM followed by LSTM or something else?

I would be happy to raise my score if the issues are addressed and there are no major other issues I have missed.

**Time Spent Reviewing:**

3 hours

---

> ### Author Response · Authors · 2021-08-11
> **Response to Reviewer 9VUm**
>
> # Description and terminology:
> As suggested, our next revision will clarify and provide more concrete definitions of terminology.  We will revise the title to "Online Meta-Learning via Learning with Layer-Distributed Memory".  We will also clarify in the introduction that LSTM hidden states/activations are referred to as memory, and that by distributed memory we mean stacked LSTMs.  We will add to the abstract that we use a deep variant of Convolutional LSTMs, trained by backpropagating through a sequence of tasks.
>
> # Longer task time spans:
> To underscore the scalability of our model, we have run additional experiments extending our task length to 300 time steps.  We train our model to solve a 10-shot 30-subtask continual learning problem  $\mathcal{T}$=($\mathcal{T}$1, $\mathcal{T}$2, $\cdots$, $\mathcal{T}$30), with each task $\mathcal{T}$i involving 1 class concept and 10 training samples for that class concept. Our model achieves an overall average performance of  70.95%, when provided with queries from each of the 30 subtasks.  In the below table, we provide the accuracy distribution on the 10-way 30-subtask continual learning problem, for every third task.
>
>
> | Model | Task 1     |Task 3     |Task 6     |Task 9     |Task 12     |Task 15     |Task 18     |Task 21     |Task 24     |Task 27     |Task 30     |
> |----------|-------------|-------------|-------------|-------------|--------------|---------------|---------------|--------------|--------------|---------------|--------------|
> |  CL+LSTM| 70.42  | 70.34  | 70.86      | 70.31       | 68.64      |69.46         | 73.32        | 70. 30     | 70.44         | 69.39       | 69.18|
>
> From this table, we see that adaptation via the hidden states is robust to catastrophic forgetting, as the performance of earlier tasks is similar to that of later tasks.  Note that, due to time constraints, we were only able to train this model for 50\% of our standard training budget; we expect additional training to yield higher overall accuracy.
>
> Furthermore, in Figure 4 (paper), we show that even on a 5-shot 20-subtask continual learning problem, gradient-based methods exhibit a significant amount of forgetting.  Our CL+LSTM model achieves a 2\% improvement in overall accuracy when compared to ANML and OML.  Thus,
> 1) We demonstrate scaling to problems with longer time spans (10-shot 30-subtask continual learning task), and
> 2) We show that, when robustness to catastrophic forgetting is crucial, gradient-based methods do not scale well for tasks with longer time steps (Figure 4 in paper).
>
> ***Guez et al:*** Indeed, they use hidden states to adapt on RL tasks, providing additional context for the broad generality of memory-based approaches.  We will add discussion of their work.
>
> # Clarifications on continual learning experiments:
>
>
> 1) **line 202:** We apologize for the ambiguity; we do not increase the number of samples, but rather increase the number of subtasks, so from $\mathcal{T}$ = ($\mathcal{T}$1, $\mathcal{T}$2, $\mathcal{T}$3, $\mathcal{T}$4,$\mathcal{T}$5), we increase the continual learning task size to $\mathcal{T}$ = ($\mathcal{T}$1, $\mathcal{T}$2, $\mathcal{T}$3,$\mathcal{T}$4, $\cdots$, $\mathcal{T}$20), making the task length 100 time steps (since each task has 5 samples).  Since each task is a class concept in itself, we are effectively increasing the number of classes in our CL-task, so this decreases the overall average performance.
>
> 2) **line 247:** Yes, every query starts with the same state.
>
> 3) **Figure 4:** As mentioned in the clarification for **line 202**, we increase the number of subtasks in our continual learning task.  We plot the task-specific accuracies when trained on a continual learning task with 20 tasks, so the x-axis is task id, and the y-axis is the accuracy.  Yes, the query set contains samples from each of the 20 subtasks.
>
> 4) **Increased accuracy with increased sample size:** In the below table, we run a 5 subtask continual learning problem with 3, 5, and 7 samples per task and observe that the accuracies increase with increased samples.  This is particularly interesting since the longer task sequence does not cause forgetting.  Moreover, the model benefits from increased samples, highlighting the effectiveness of adaptation via hidden states for continual learning tasks.
>
> | Model | 3-shot 5-subtask Continual Task     |5-shot 5-subtask     |7-shot 5-subtask      |
> |----------|-----------------------------------------------|---------------------------|---------------------------|
> |  CL+LSTM| 98.03 | 98.31  | 98.75   |
>
> # Clarification on Section 4.5:
> Our motivation here is to demonstrate that ConvLSTM models can be deployed when the nature of the task is unknown (e.g., whether it is few-shot or standard supervised learning).  So, we treat the entire supervised learning task as one big task and never reset the hidden states.  However, we update the model via back propagation. CL-ResNet-20 essentially is a ResNet-20 with each convolutional layer swapped out by a convolutional LSTM layer.  The naming convention for CL-ResNet is similar to that of CL-UNet as mentioned in **line 295**.

---

### Author Response · Authors · 2021-08-11
**Joint response to all reviewers**

We thank the reviewers for valuable feedback and suggestions.  As requested, we will clarify several elements of the presentation in the next revision.  We will also add citation to and discussion of additional references pointed out by reviewers.  Details of these changes are described in individual responses below.


**On motivation:**
Using memory allows the network to implicitly encode a learning algorithm for task adaptation, as opposed to most other meta-learning methods wherein the adaptation algorithm is fixed (Example:OML-uses gradient descent; CPM-uses weighted nearest neighbor over prototypes).  We elaborate on the wide applicability, efficacy, and significance of such memory-based adaptation methods in specific responses below.


**On datasets:**
Some reviewers have raised concerns over the datasets used and cite datasets used in standard few-shot learning.  While we agree that, in standard few-shot learning, Omniglot and CIFAR have reached saturation, in online few-shot learning these datasets have not reached saturation and there exists a significant gap in performance between the offline and online setups for these datasets.  See below for additional details.

**On scalability:**
We understand the questions over scalabilty of our models and provide additional experiments to alleviate concerns. We add semantic segmentation experiments on higher resolution images ($112 \times 112$) and deeper networks (16 layers).  Moreover, to ameliorate concerns over ability to learn over longer time scales, we add a 300 time-step 10-shot 30-subtask continual learning experiment, and show that the model exhibits no forgetting.  In fact, results show that with longer time steps, our model improves its performance due to the additional samples provided; we thus also scale well with data size.

---

### Decision · Program_Chairs · 2021-09-27

**Decision:**

Accept (Poster)

**Comment:**

The paper operates in a simple approach to meta-learning, of running a recurrent network through tasks and back-propagating to learn the adaptation. They find that simple deep-convolutional lstm networks (with details) outperforms many of the previous architectures studies before on standard benchmarks and interestingly also works in classic supervised setting at state of the art level (as a single system).

On the positive side, the method is simple and the experiments are done carefully. The major drawback is that it is not being tested on harder problems and more modern online learning settings. Pushing the system this would we provide a major improvement to the paper. Despite this, the currently results are still sufficiently interesting. The are few points in terms of writing the paper that should be improved and I hope the authors will address that for the final version.